# Targeting Fat Oxidation in Mouse Prostate Cancer Decreases Tumor Growth and Stimulates Anti-Cancer Immunity

**DOI:** 10.3390/ijms21249660

**Published:** 2020-12-18

**Authors:** Amanda Guth, Emily Monk, Rajesh Agarwal, Bryan C. Bergman, Karin A. Zemski-Berry, Angela Minic, Kimberly Jordan, Isabel R. Schlaepfer

**Affiliations:** 1Department of Clinical Sciences, Flint Animal Cancer Center, Colorado State University, Fort Collins, CO 80523, USA; amanda.guth@gmail.com; 2Division of Medical Oncology, University of Colorado Anschutz Medical Campus, Aurora, CO 80045, USA; emily.monk@cuanschutz.edu; 3School of Pharmacy, University of Colorado Anschutz Medical Campus, Aurora, CO 80045, USA; rajesh.agarwal@cuanschutz.edu; 4Division of Endocrinology, Metabolism and Diabetes, University of Colorado Anschutz Medical Campus, Aurora, CO 80045, USA; bryan.bergman@cuanschutz.edu (B.C.B.); karin.zemskiberry@cuanschutz.edu (K.A.Z.-B.); 5Department of Immunology and Microbiology, University of Colorado Anschutz Medical Campus, Aurora, CO 80045, USA; angela.minic@cuanschutz.edu (A.M.); kimberly.jordan@cuanschutz.edu (K.J.)

**Keywords:** CPT1A, prostate cancer, acyl-carnitines, ranolazine, CD8 T-cells, dendritic cells, lipid metabolism

## Abstract

Lipid catabolism represents an Achilles heel in prostate cancer (PCa) that can be exploited for therapy. CPT1A regulates the entry of fatty acids into the mitochondria for beta-oxidation and its inhibition has been shown to decrease PCa growth. In this study, we examined the pharmacological blockade of lipid oxidation with ranolazine in TRAMPC1 PCa models. Oral administration of ranolazine (100 mg/Kg for 21 days) resulted in decreased tumor CD8^+^ T-cells Tim3 content, increased macrophages, and decreased blood myeloid immunosuppressive monocytes. Using multispectral staining, drug treatments increased infiltration of CD8^+^ T-cells and dendritic cells compared to vehicle. Functional studies with spleen cells of drug-treated tumors co-cultured with TRAMPC1 cells showed increased ex vivo T-cell cytotoxic activity, suggesting an anti-tumoral response. Lastly, a decrease in CD4^+^ and CD8^+^ T-cells expressing PD1 was observed when exhausted spleen cells were incubated with TRAMPC1 Cpt1a-KD compared to the control cells. These data indicated that genetically blocking the ability of the tumor cells to oxidize lipid can change the activation status of the neighboring T-cells. This study provides new knowledge of the role of lipid catabolism in the intercommunication of tumor and immune cells, which can be extrapolated to other cancers with high CPT1A expression.

## 1. Introduction

Prostate cancer (PCa) is a multifactorial and heterogenous disease and one of the most diagnosed malignancies in the world [1,2]. PCa exhibits a unique metabolism that changes from initial diagnosis to aggressive PCa and metastasis. In fact, identification of PCa metabolic subclasses with different clinical outcomes could be predicted from metabolic expression profiles of tumors, underscoring the link between deregulated lipid metabolism and PCa outcomes [3]. Studies have shown that excess body weight, periprostatic adipose tissue, and circulating lipids can act as promoters of PCa progression and invasion [4,5]. These observations are not specific to PCa as excess lipid and energy is known to establish a tumor environment with features of endocrine resistance and metabolic dysfunction [6]. The ways hormone-dependent cancers use lipids from the environment is not exactly known but beta oxidation of lipids in the mitochondria appears to be a dominant bioenergetic pathway in PCa [7,8]. Indeed, the availability of long-chain fatty acids to PCa cells can increase the activity of the rate-limiting enzyme for beta-oxidation, carnitine palmitoyl transferase 1 (CPT1) [9]. Our previous studies focused on the CPT1A isoform and its role in PCa, promoting growth, glycolysis changes, and drug therapy resistance [8,10,11]. However, the role of CPT1A and beta oxidation in the immune environment of PCa remains poorly defined.

When the patient’s immune system is not able to eliminate malignant cells, tumors progress to a cancer-induced immunosuppressive status. Efforts to re-engage the immune system to attack the tumors is an area of intense investigation. In PCa, the autologous cellular immunotherapy Sipuleucel-T is the first therapeutic vaccine approved for PCa patients [12,13]. This cellular approach is manufactured from antigen presenting cells (dendritic cells) primed to recognize the prostate acid phosphatase. Reinfusion of these modified cells back in the patient generates an antigen-specific T-cell immune stimulation. However, this therapy does not improve progression-free survival. It prolongs median overall survival by 4 months compared to placebo treatments. Thus, finding new ways to stimulate the immune response in PCa is a critical need and a promising area of new research.

The essential role of the patient’s immune system in fighting cancer has led to the discovery of immune molecules, including vaccines and immune checkpoint protein (ICP) inhibitors like anti-CTLA-4 and anti-PD1/PDL1 monoclonal antibodies [14]. This efficient strategy of checkpoint blockade represents one of the main oncological breakthroughs, with remarkable clinical durable responses and survival advantages observed in several cancer types [15], but not in PCa. This could be because advanced PCa shows significantly less infiltration of T-cells as compared to benign prostatic hyperplasia [16]. These observations indicate that PCa progression could be associated with defects in the cell-mediated immune responses. Additionally, the low mutational load of PCa can contribute to the failure of ICP, thus finding alternative, safe ways to enhance the immune response towards PCa is novel and worth pursuing.

Distinct metabolic programs are initiated upon T-cell activation, differentiation, and effector and memory transitions in lymphocytes. This metabolic reprogramming can be altered by nutrients in the environment and/or ICP (e.g., PD1, CTLA-4) on the cell surface, limiting effector T-cell differentiation and function [17,18]. Mechanistic understanding of the impaired function of T-cells in tumors is still limited but recent insights from using metabolic inhibitors in hormone-dependent cancers offer insights on the role of lipids in the reactivation of the immune system. For example, Carpenter et al. showed that blocking Liver-X-Receptors (LXR) induced antitumor immunity by stimulation of the CD8^+^ T-cell cytotoxic activity and mitochondrial metabolism in aggressive breast cancer [19]. Another study suggested that T-cells use glycolysis to expand and activate, while lipid oxidation keeps them in a suppressed state, unable to attack tumors [20]. In fact, CPT1A is the mediator of the increased lipid oxidation in T-cells exposed to the PD1 checkpoint inhibition, strongly suggesting that metabolic reprograming of the T-cells in tumors is a promising approach that could be used to re-activate the immune system and invigorate anti-tumor immunity.

The use of clinically approved lipid metabolic inhibitors can help facilitate translational studies [21,22]. Our group has studied the genetic and pharmacological targeting of CPT1A using etomoxir and clinically used drugs like perhexiline and ranolazine, making our studies more translational [23,24]. The partial lipid oxidation inhibitor ranolazine is FDA-approved for the treatment of chronic stable angina [25]. It mainly works by inhibiting late sodium currents and increasing adenosine in the heart [26,27]. Although it is a weak inhibitor of fat oxidation in benign tissues, our work and others have found that it decreases cancer cell viability [28,29,30]. Etomoxir and perhexiline are potent inhibitors of CPT1 enzymes; although etomoxir is not approved for human use, it serves as a great tool in the lab to study fat oxidation in onco-immunology [31]. Currently, there are few studies of these metabolic inhibitors and their role in the immune system in PCa, making them excellent tools to study how lipid oxidation modulates the immune system. In this study, we tested the ability of systemic ranolazine to block tumor growth and stimulate the immune response in TRAMPC1 allografts in immunocompetent mice.

## 2. Results

### 2.1. Systemic Treatment with Ranolazine Decreases Tumor Growth in Immune Competent Mice

Earlier studies showed that ranolazine treatment can reduce tumor growth in hormone-dependent cancers [29,30]. However, the role of the immune system was not considered in these immune-deficient nude mouse studies. The TRansgenic Adenocarcinoma of Mouse Prostate (TRAMP) model is a transgenic line of C57BL/6 mice that harbors the SV40 large T antigen driven by the rat probasin promoter [32]. The development and progression of PCa in this model mimics human disease, including a lipid phenotype that expresses CPT1A [33]. The TRAMPC1 cell line was derived from a PCa tumor from a 32-week-old TRAMP mouse, and it easily forms tumors in syngeneic mice [32]. Additionally, TRAMPC1 cells have been previously used in cytotoxic assays when co-cultured with natural killer cells exposed to the chemopreventive agent sulforaphane [34]. Thus, we used TRAMPC1-derived allografts in C57Bl mice to test if the immune system would promote or hinder the effect of ranolazine on tumor growth. Figure 1B shows results of the tumor growth over 4 weeks at two different doses (40 and 60 mg/Kg) in mice treated 3 times a week (excluding weekends), two-way ANOVA; *p* < 0.0001 for the time-drug interaction. Since we observed a dose effect on growth, we repeated the experiment with a 100 mg/Kg for 3 weeks in another set of mice, Figure 1C. Significant results were observed at 15 days after the beginning of the treatment. Repeated measures (RM) ANOVA indicated significant results of the time-drug interaction (*p* < 0.0001), as well for the drug alone (*p* = 0.017). In both studies, mice did not show significant changes in body weight between treatments, Figure 1D,E. Tumors and blood from these mice were used for molecular characterization shown below.

### 2.2. Treatment with Ranolazine Results in Changes in Immune Check Point Proteins (ICP) and Macrophages in the Tumors

To characterize the tumors of treated mice, we collected fresh tumor tissue, disaggregated the cells, and processed them for flow analysis of ICP surface markers, Figure 2. We observed decreased content of percentages of CD8 T-cells staining with Tim3 ICP, while no changes were observed with expression of the PD1 and Lag3 markers. This suggests a potential change in the immune phenotype of the CD8 T-cells present in the tumors following treatment. No significant changes were observed in the CD4 T-cells, Appendix A. We also observed a significant increase in the percentage of tumor macrophages in the tumors with the drug treatment (1.5-fold, *p* = 0.03), suggesting potential recruitment of phagocytic cells to reduce tumor burden, Figure 2C. Lastly, regarding blood monocytes, the drug treatment resulted in less inflammatory monocytes (i-mono) with PDL1 stain by mean fluorescence intensity (MFI) only (*p* = 0.01 vs. vehicle), Figure 2E.

### 2.3. Increased Content of CD8 T-Cells and Dendritic Cells in Drug-Treated Tumors

We next used multispectral fluorescence immunohistochemistry to localize and quantify immune cells in tumor sections [35]. Figure 3 shows multispectral fluorescence (Vectra 3) images obtained from the TRAMPC1 tumors, showing higher infiltration of CD8 cells into the tumors treated with ranolazine. Images showed the infiltration of the immune cells into the tumors, as well as their proximity to the TRAMPC1 tumor cells Figure 3A. Quantification of the images using the Inform software indicated that CD8^+^/CD3^+^ cells were increased by ~3-fold with the drug treatment compared to vehicle, Figure 3B,C. Interestingly, PD1 expression was also significantly increased by 2-fold in the drug-treated tumors compared to vehicle, Figure 3D. These staining results seem discrepant with the flow data, however, the ratio of CD8^+^PD1^+^ to CD8^+^ was 27% for the drug-treated tumors compared to 40% for the vehicle-treated tumors. Overall, PD1^+^ stain was not significantly changed between vehicle and drug treatment, Appendix A. Another possibility is that there might be more focal pockets of T-cells being activated and expressing PD1 in the intact tumors compared to the overall flow cytometry analysis. Dendritic cells were also significantly increased in the drug-treated tumors compared to vehicle, Figure 3E, and this was not accompanied by increased PD1 expression (Figure 3F), suggesting that more active antigen presentation may be occurring in the drug-treated tumors.

### 2.4. Increased Ex Vivo T-Cell Cytotoxic Activity of Drug-Treated Spleens

To investigate if spleen-derived cells were able to elicit an immune response against the tumor, we used an ex-vivo assay using real-time quantification of cells undergoing caspase-3/7 mediated apoptosis using the Incucyte live-imaging system. Briefly, red-labeled TRAMPC1 tumor cells were mixed with T-cells derived from tumor-bearing mouse spleens at different ratios to assess the cytotoxic activity against the tumor cells, Figure 4. Ten replicates were used per condition. The 1:5 ratio was also tested but did not produce meaningful results (data not shown). We observed a significant increase in cytolytic activity from drug-treated spleen cells compared to vehicle-treated cells, using the 1:25 ratio (42% change in means between groups, *p* = 0.0012). The 1:10 ratio was trending but not significantly changed between drug- and vehicle-treated tumor cells (11.2% change between groups, *p* = 0.07). These results indicated that the systemic drug treatment could increase the cytotoxic activity of T-cells against the tumor.

### 2.5. Mouse Prostate Cancer Cells Burn Less Lipid and Generate Less Acyl-Carnitines When Exposed to Ranolazine

From the apoptotic results above, we reasoned that a defect in tumor fat oxidation could be underlying the differences in T-cell cytotoxic activity. We next studied the capacity of the TRAMPC1 cells to burn fatty acids commonly found in blood (like oleic acid) in response to ranolazine treatments using the seahorse flux analyzer, Figure 5A. Treatment of tumor cells with 75 µM ranolazine resulted in decreased maximal respiration rate, compared to vehicle, when cells were incubated with 25 µM of oleic acid as their only source of carbon supply, (*p* < 0.001). Additionally, analysis of the production of long-chain acyl carnitines (the product of CPT1 activity) was also significantly decreased in the TRAMPC1 cells treated with 75 µM ranolazine for 48 h (20% less, *p* = 0.033), Figure 5B. Parallel results were also observed in MyC-CaP mouse PCa cells (40% less, *p* = 0.028), which is another model of aggressive PCa that is androgen-independent, Figure 5C.

### 2.6. Decreased PD1 Stain in T-Cells Incubated with TRAMPC1 Cells Deficient in Cpt1A Expression

To study the role of Cpt1A and lipid oxidation in immune cells, we used a lentivirus approach to knock down (KD) Cpt1A expression in the TRAMPC1 cells. We used three different shRNAs and selected sh2 to proceed with the experiments, as it produced the best KD results, Figure 6A. This selected Cpt1a-KD cell line was able to decrease basal (50%) and maximal respiration measurements when the cells were challenged with 25 µM oleic acid in a seahorse oxygen consumption rate (OCR) assay, Figure 6B. Next, we studied the effect of co-culturing the Cpt1a-KD cells with isolated spleen cells from seven mice (four male + three female), Figure 6C,D. A decrease in the percentage of CD4^+^ T-cells expressing PD1 and CD8^+^ T-cells expressing PD1 was observed when the exhausted spleen cells were incubated with the Cpt1a-KD cells compared to the control cells (NTsh cell line). These changes were not associated with decreased viability of the T-cells, Appendix A. These data indicated that genetically blocking the ability of the tumor cells to burn lipid can change the activation status of the neighboring T-cells.

## 3. Discussion

While improvements in immunology have led to the identification of immune checkpoint proteins like PD1 that trigger reduced T-cell anti-tumor activity, these methods do not work well in PCa patients. Thus, there is a critical need to devise novel, safe ways to stimulate the immune system to attack tumors. Currently there are few studies looking at the role of CPT1A-mediated lipid metabolism in infiltrated T-cells in PCa tumors [31,36]. A recent study identified upregulation of Cpt1a in tumors and in CD8^+^ T-cells of obese mice, but no PCa tumors nor Cpt1a inhibitors were examined in that study [37]. Another paper showed that increased Cpt1a in liver tumors induced CD4 T-cell apoptosis and this could be reversed with systemic perhexiline treatment [38], which is a clinically used inhibitor of the carnitine shuttle in cardiology.

In this study, we provide evidence that targeting lipid oxidation in mice with the partial fat oxidation inhibitor ranolazine results in decreased growth and immune activation against the tumor. The safety of the oral ranolazine treatments could be seen in our results with increasing drug doses and lack of animal toxicity, while significantly decreasing tumor burden. What remains unknown is if the duration of this anti-tumor effect persists after discontinuing the drug treatment. Molecular examination of tumor samples at the end of the study revealed increased infiltration of macrophages, CD8^+^ cells, and dendritic cells in the drug-treated tumors. This suggests an important role for lipid beta-oxidation in the tumor microenvironment (TME) and the anti-tumor immune response.

The TME is characterized by nutrient competition, low pH, and limited oxygen, resulting in an immunosuppressive environment that allows tumor growth. This starvation-like tolerogenic environment relies more on fat oxidation to fulfill the energy needs of all the cells, promoting differentiation of T-regs and macrophages that accelerate T-effector cell exhaustion [39]. In our studies, we have found that the systemic use of ranolazine over a month changed the TME and the tumor cell’s ability to activate the immune cells. Particularly, we found that blockade with ranolazine decreased checkpoint inhibitors in CD8 cells (Tim3, Figure 2) and increased the number of infiltrated macrophages, suggesting a less tolerogenic environment associated with smaller tumors. Recent studies have also shown that a type of immune-suppressive macrophages called M2 rely more on the Krebs cycle and fat oxidation compared to the more inflammatory M1 counterparts [40]. Thus, it is possible that by reducing the fat oxidation capacity of the infiltrated macrophages in our treated tumors, a metabolic reprograming towards a more immunogenic phenotype was achieved.

Dendritic cells were also increased in the drug-treated tumors. Interestingly, fat oxidation also seems to drive these cells towards a tolerogenic phenotype, showing high mitochondrial oxidative capacity [41]. Inhibition of fat oxidation of the dendritic cells with etomoxir resulted in expression of maturation factors and a modest increase in T-cell stimulatory function [41]. Thus, fat oxidation in dendritic cells of the TME could be modulated with pharmacological interventions to stimulate T-cell anti-tumor activity. In fact, our studies of tumor cells co-cultured with spleen cells from drug-treated mice showed increased apoptosis of the tumor cells. This suggests that the drug-treated immune cells can produce metabolites or cytokines that could modify the TME and its immune constituents to become less tolerogenic.

From our analysis of acyl carnitines in TRAMPC1 and MyC-CaP cells, it is reasonable to speculate that fatty acid-carnitine derivatives produced by the tumor cells could be supporting the fat oxidation phenotype of the tolerogenic TME, supporting cancer growth.

To address the concerns that ranolazine has other effects besides partial fat oxidation inhibition, we used a knockdown approach to reduce Cpt1a expression in TRAMPC1 cells. The decrease in fat oxidation resulted in less PD1 expression in T-cells (Figure 6), an effect that has been shown to rescue the effector function of tumor-infiltrating lymphocytes by restoring glycolysis [20,42]. The possibility that checkpoint inhibitors also play a role in other immune cells like myeloid cells remains to be explored [43].

This study provides new knowledge of the role of lipid catabolism in the intercommunication of tumor and immune cells, which can be extrapolated to other cancers with high CPT1A expression like breast, ovarian, and colon cancers. Treatment of patients with CPT1A inhibitors could potentially re-activate the immune cells to restore and enhance cellular-mediated antitumor immunity and tumor regression.

## 4. Materials and Methods

### 4.1. Cell Lines and Reagents

The mouse PCa TRAMPC1 cells (Gift from Dr. Agarwal, 2015, originally from ATCC) were grown in DMEM with 4 mmol/L L-glutamine, 5 μg/mL insulin, 10 nmol/L dehydrotestosterone (DHT), 5% FBS, and 5% Nu Serum (Thermo Fisher Scientific, Walthman, MA, USA). Ranolazine (Sigma-Aldrich, St. Louis, MO, USA) was prepared as a 40 mM stock solution in PBS and stored at −20 °C. MyC-CaP cells (Gift from Dr. Owens, 2018) were grown in RPMI media containing 5% FBS supplemented with amino acids and Insulin (Thermo Fisher). Knockdown cell lines were made using Sigma’s MISSION shRNA available through the University of Colorado Functional genomics core.

### 4.2. Mouse Xenograft Generation with TRAMPC1 Cells

All mouse studies, including blood and tissue collection, were approved by the Institutional Animal Care and Use Committee (IACUC) of the University of Colorado (protocol number 00314; approved 12-16-2019). Six-to-eight-week-old male mice (C57Bl/J6) were purchased from Charles Rivers Labs (Wilmington, MA, USA). Mice were housed in regular ventilated mouse cages with ad libitum access to water and regular mouse chow. Their flanks were shaved before cell inoculations under the skin with TRAMPC1 parental cells to generate the xenografts. About 1 × 10^6^ cells were injected per flank, and tumor growth was monitored three times a week with calipers. When tumors were palpable, we initiated treatment by oral gavage with ranolazine at (40, 60 or 100 mg/kg) every other day for about three weeks. Tumor growth was calculated using the formula volume = (π × length × width^2^)/6.

### 4.3. Tissue Harvesting and Flow Cytometry Assays

At the completion of the study, 21 days after start of treatment or when total tumor burden reached the protocol limits, the animals were euthanized by CO_2_ inhalation and cervical dislocation, followed by blood collection and tissue harvesting. Blood was collected via cardiac puncture and used fresh for flow analysis. About 50 mg of spleen and tumor tissue were collected and digested into single-cell suspensions as described previously [44]. A small portion of each tumor was formalin-fixed for further histological analyses. Blood and spleen-derived samples were incubated with fluorescently labeled antibodies to evaluate the following markers: mouse leukocytes (mouse CD45, CD3, CD4, CD8, CD19, CD20), inhibitory receptors (PD1, Tim3, CTLA-4), and myeloid cell subsets (CD33, CD11b). Isotope control antibodies were used to establish gating parameters. All antibodies used were purchased from Thermo Fisher (Waltham, MA, USA). Samples were run on a Gallios Flow Cytometer (Beckman Coulter, Miami, FL, USA) and data analyzed using FlowJo software (Becton Dickinson, San Jose, CA, USA).

### 4.4. Multispectral Fluorescence Immunohistochemistry

Tumor tissue was fixed in formalin and paraffin-embedded for multispectral imaging on the Vectra 3.0 Automated Quantitative Pathology Imaging System (Akoya Biosciences, Menlo Park, CA, USA). Tissue sections of 5 μm were mounted on glass slides and sequentially stained using a Bond Rx Autostainer (Leica, Buffalo Grove, IL, USA) for the following mouse panel: DEC205 (Biorad, Hercules, CA, USA), PD1, CD11c (Cell Signaling Technologies, Danvers, MA, USA), CD3 (Cell Signaling Technologies), CD8 (Cell Signaling Technologies), and ECadherin (Agilent, Santa Clara, CA, USA). Slides were dewaxed and heat-treated in epitope retrieval solution 2 (Leica), blocked in antibody (Ab) diluent (Akoya Biosciences), incubated for 30 min with the primary antibody, 10 min with horseradish peroxidase-conjugated anti-rabbit secondary polymer (Akoya Biosciences), and 10 min with horseradish peroxidase-reactive OPAL fluorescent reagents (Akoya Biosciences). Slides were washed between staining steps with Bond Wash (Leica) and stripped between each round of staining with heat treatment in antigen retrieval buffer. After the final staining round, the slides were stained with spectral 4′,6-diamidino-2-phenylindole (DAPI, Akoya Biosciences), and cover slipped with Prolong Diamond mounting media (Thermo Fisher). Whole slide scans were collected using the 10× objective at a resolution of 1.0 μm. Using Phenochart (Akoya Biosciences) software, ten regions of interest were randomly chosen for all tumors followed by multispectral imaging with the 20× objective at a resolution of 0.5 μm. The multispectral images were analyzed with inForm software (Akoya Biosciences) to unmix adjacent fluorochromes; subtract autofluorescence; segment the tissue into tumor regions and stroma; segment the cells into nuclear, cytoplasmic, and membrane compartments; and to phenotype the cells according to cell marker expression. A summary of cell phenotypes and marker expression for each slide was generated with the phenoptrReports feature of the Akoya Biosciences’ Inform software.

### 4.5. Co-Culture Studies of TRAMPC1 Cells with Drug-Treated Derived Splenocytes

Red-labeled TRAMPC1 cells (NucRed-treated, R37106, ThermoFisher, Waltham, MA, USA) were plated at a density of 1000 per well in a 96-well plate overnight. The next day, spleen cells isolated from three TRAMPC1-tumor bearing C57Bl mice, either untreated or treated with 100 mg/kg ranolazine, were thawed and counted. Media was removed from the 96-well plate. Spleen cells were added to each well in a 150 μL volume at the following ratios: for 1 to 5, 5000 spleen cells per well were added. For 1 to 10 ratio, 10,000 cells were added and for 1 to 25 ratio, 25,000 cells were added. Lastly, 50 μL of the Caspase-3/7 Green Reagent for Apoptosis (Essen BioScience, Ann Arbor, MI, USA) was added at the appropriate final concentration according to the manufacturer’s instructions. Plates were then placed in the Incucyte^®^ live imaging incubator (Essen BioScience) for 5 days and imaged every 3 h. GraphPad Prism v8 (GraphPad Software, La Jolla, CA, USA) was used to analyze the data. The following variables were used: phase contrast; percentage of well that is covered by cells. Green phase: percentage of well that is covered by green cells. Green cell count: number of green cells per mm^2^.

### 4.6. Lentiviral shRNA Transfections

Knockdown cell lines were made using lentiviral particles prepared in the CU Functional Genomics core. The following shRNAs were used: TRCN0000110597, TRCN0000110598, TRCN0000110599, and control (SHC016). Lentiviral particles were generated at the Functional Genomics Core at the University of Colorado (Aurora, CO, USA). Lentiviral transduction and selection were performed according to Sigma’s MISSION protocol. Puromycin (2 μg/mL) Sigma (St. Louis, MO, USA) was used for cell line drug selection.

### 4.7. Co-Culture Studies of Cpt1A-KD Cells with Splenocytes

For the co-culture studies with control (NTsh) and Cpt1A-KD cells, spleens were freely obtained from unused C57B/6 mice (originally purchased from Jackson Laboratories) at Colorado State University under approved IACUC protocols. Four male and three female C57Bl/J6 spleens were pooled and used for isolation of fresh spleen cells for the studies.

### 4.8. Western Blot Analysis

Protein extracts of 20 µg were separated on a 4–20% SDS-PAGE gel and transferred to nitrocellulose membranes as described [10]. Band signals were visualized with the LICOR system (LI-COR Biosciences, Lincoln, NE, USA). Antibodies: CPT1A: Protein Tech 15184-1-AP, Rosemont, IL, USA; GAPDH: Cell Signaling Technologies 5174.

### 4.9. Acyl Carnitine Analysis

Cells were homogenized in 500 µL water and an aliquot (20 µL) taken for protein concentration. The homogenized sample was brought up to 750 µL with water and then methanol (900 µL) was added. Samples were then processed at the CU Nutrition Obesity Research Center (NORC) Lipidomics Core Facility. Briefly, mass spectrometric analysis was performed in the positive ion mode using multiple-reaction monitoring (MRM) of ten acylcarnitine molecular species and the two deuterated internal standards. The precursor ions monitored were the molecular ions [M + H]^+^ and the product ion for all acylcarnitine species was at *m*/*z* 85 corresponding to loss of fatty acid and trimethylamine [45]. Quantitation was performed using stable isotope dilution with a standard curve for long-chain and short-chain acylcarnitines and results were normalized to protein content. Full description of the protocol is in Appendix A.

### 4.10. Seahorse Metabolic Flux Analysis

Mitochondrial respiration measurements were done at the University of Colorado molecular and cellular analytical core, using a Seahorse XFe96 Analyzer with 96-well plates (Agilent, Santa Clara, CA, USA). Cells (40,000/well) were grown in for 48 h as detailed above. Spent media was then aspirated, cells washed with PBS, then supplemented with Agilent Seahorse XF RPMI media pH 7.4 (Cat No. 103576-100) the night before the assay. The next day, cells were supplemented with 25 μM oleic acid conjugated to albumin (Sigma), 500 μM L-carnitine (Sigma), plus vehicle (PBS) or ranolazine two hours before the assay. Oxygen consumption rate (OCR) was analyzed using the XF Cell Mito Stress test, which provides a standard method to assess mitochondrial function in live cells. This test uses four chemical inhibitors: Oligomycin (2 μM), FCCP ((4-(trifluoromethoxy) phenyl) carbonohydrazonoyl dicyanide, 2.5 μM), and a mix of rotenone (125 nM) and antimycin A (5 μM) to modulate the electron transport chain. All these drugs were purchased from Sigma and their stocks stored at −80 °C. 

### 4.11. Statistics

Statistical analysis was carried out with GraphPad Prism software v8, GraphPd Software (La Jolla, CA, USA) Comparisons between groups were done with Student *t*-tests, Mann–Whitney tests, or ANOVA tests, followed by post hoc tests when appropriate, considering alpha = 0.05. All data represent mean ± SD unless otherwise indicated.

## 5. Conclusions

Treatment of cancer patients with lipid metabolic inhibitors like ranolazine could potentially re-activate the immune cells to restore and enhance cellular-mediated antitumor immunity and tumor regression.

## Figures and Tables

**Figure 1 ijms-21-09660-f001:**
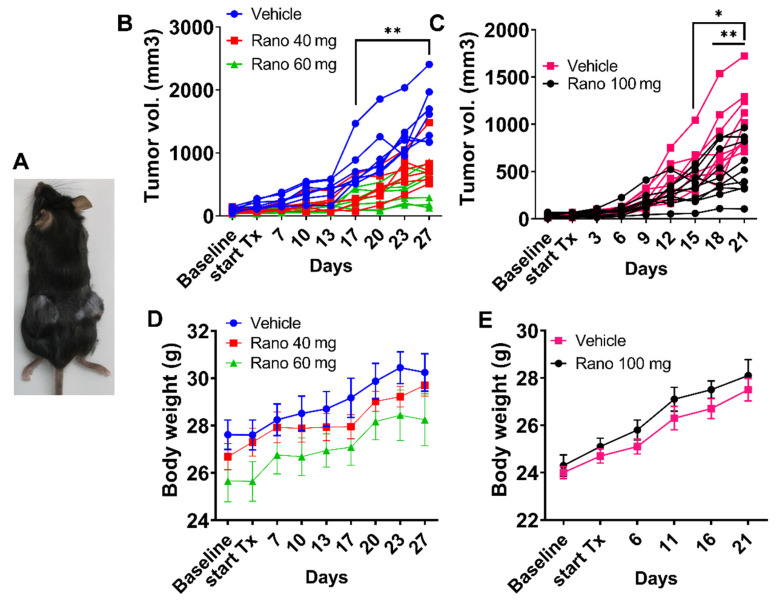
Systemic treatment with ranolazine decreases tumor growth in immune competent mice. (**A**) C57BL mice with syngeneic allografts of TRAMPC1 cells in both flanks. (**B**) Changes in tumor volume over time in mice bearing TRAMPC1 tumors and systemically treated orally with vehicle (PBS), 40 or 60 mg/kg 3 times a week of ranolazine over 4 weeks. Two-way ANOVA *p* = 0.001 (drug), ** *p* < 0.01, each treatment vs. vehicle; *n* = 7 for vehicle and 60 mg dose; *n* = 6 for 40 mg dose. (**C**) Same as in B but with 100 mg/kg over 3 weeks, repeated measures (RM)-ANOVA *p* = 0.017 (drug); * *p* = 0.03, ** *p* < 0.01 vs. vehicle, *n* = 10 per group. (**D**,**E**) Body weight of the mice (Mean ± SEM) over the course of the studies with 40 or 60 mg (**D**) or 100 mg (**E**).

**Figure 2 ijms-21-09660-f002:**
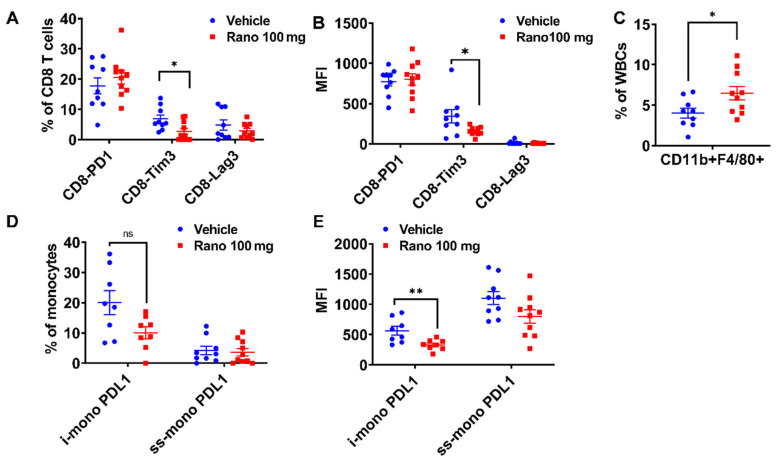
Treatment with ranolazine results in changes in immune check point proteins (ICP) and macrophages in the tumors. Tumors of treated mice (Figure 1C, 100 mg/kg) were collagenase-treated and used for flow analysis of ICP surface markers. (**A**,**B**) Mean fluorescence intensity (MFI) of PD1, Tim3, and Lag3 markers in exhausted CD4 (**A**) or CD8 (**B**) T-cells from treated tumors with 100 mg/kg ranolazine (vehicle, *n* = 9; rano *n* = 10), Mann–Whitney test; * *p* < 0.05. (**C**) Percentage of tumor macrophages in tumors treated with 100 mg/kg ranolazine; *t*-test; * *p* = 0.03. (**D**,**E**): Percentage (**D**) or MFI (**E**) of blood myeloid monocytes expressing PDL1 ligand in ranolazine-treated mice. Mann–Whitney test; ** *p* = 0.01. i-mono = inflammatory monocyte (likely immune suppressive); ss-mono = steady state monocytes.

**Figure 3 ijms-21-09660-f003:**
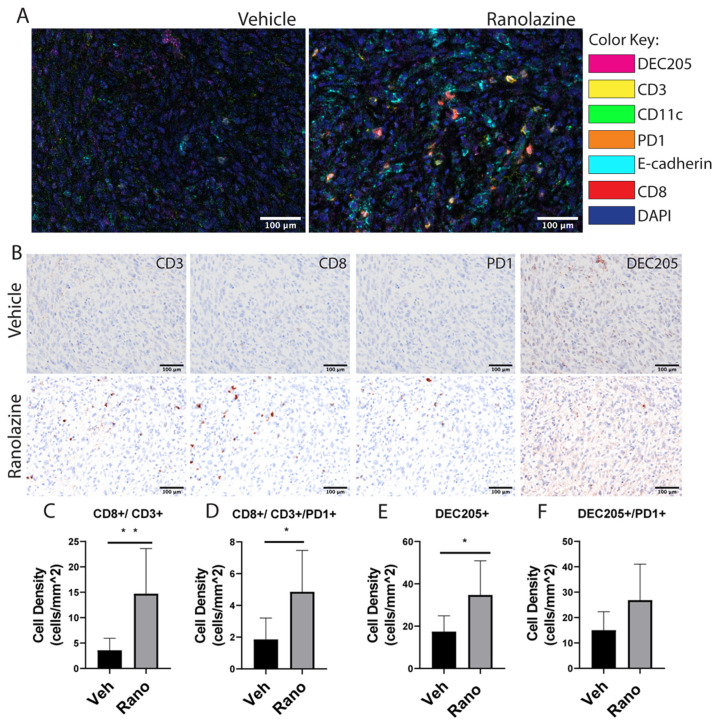
Immune cells are increased in the TRAMPC1 tumors following oral ranolazine treatment. (**A**) Tumors from C57/Bl mice treated with vehicle or ranolazine 100 mg/kg were stained with two seven-color IHC panels and analyzed by multispectral imaging. (**B**) Representative brightfield images of CD3, CD8, PD1, and DEC205 individual stains for the same cut of vehicle (top) or drug (bottom) tumors. Cellular phenotypes were analyzed using inform (Akoya) software and cell density was calculated for CD8^+^CD3^+^ (**C**), CD3^+^CD8^+^PD1^+^ (**D**), DEC205^+^ (**E**), and DEC205^+^PD1^+^ (**F**) markers. Students *t*-test, * *p* < 0.05; ** *p* < 0.01; *n* = 7 per condition.

**Figure 4 ijms-21-09660-f004:**
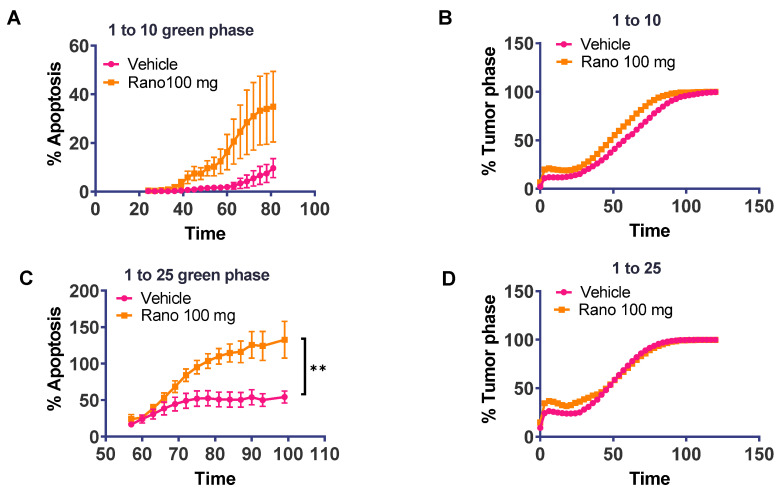
Increased ex vivo cell cytotoxic activity of drug-treated spleens; Cells from spleens of drug-treated mice were mixed with NucRed labeled TRAMPC1 cells at different ratios to assess the cytotoxic activity of the tumor cells, using a 96-well plate fluorescent caspase assay. (**A**) Percentage of apoptosis using a 1:10 ratio of TRAMPC1 to spleen cells. (**B**) Corresponding tumor phase to show growth of tumor cells with both conditions. (**C**,**D**) Same as in A but at 1:25 ratio. Two-way ANOVA: ** *p* = 0.0012 (42% difference in green counts between vehicle and ranolazine treatment). Data represents Mean + SEM; *n* = 10 per condition.

**Figure 5 ijms-21-09660-f005:**
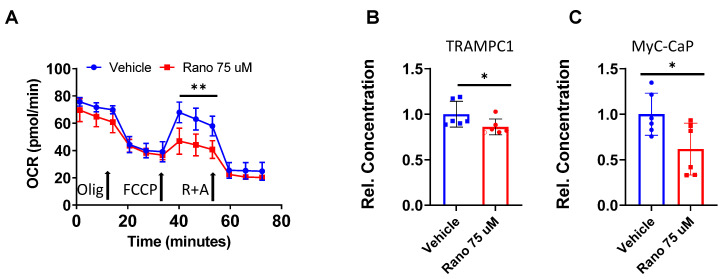
Mouse prostate cancer cells burn less lipid and generate less acyl-carnitines when exposed to ranolazine. (**A**) Decreased oxygen consumption rate (OCR) in TRAMPC1 cells treated with ranolazine and 25 µM of oleic acid conjugated to BSA for 2 h prior to the assay. The significant section of increased OCR corresponds to the maximal respiration after addition of FCCP. Holm–Sidak multiple comparisons *t*-test: ** *p* < 0.001 vs. vehicle. Olig = oligomycin; R + A = rotenone + antimycin A. (**B**,**C**) Decreased acyl-carnitine production after treatment with 75 µM ranolazine for 24 h in TRAMPC1 (**B**) and MycCaP (**C**) cells, *t*-test: * *p* < 0.05 vs. vehicle.

**Figure 6 ijms-21-09660-f006:**
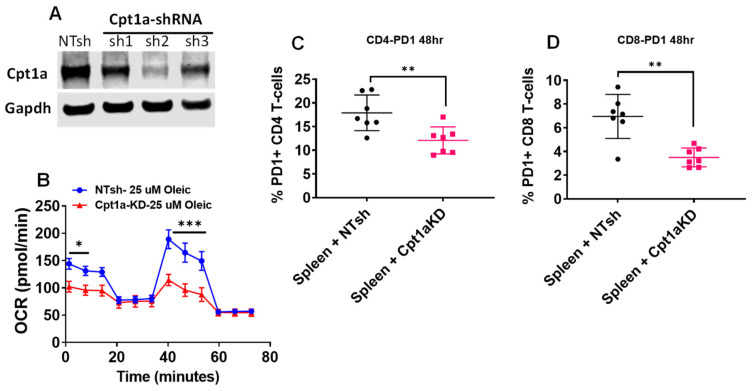
Decreased PD1 stain in T-cells incubated with TRAMPC1 cells deficient in Cpt1a expression: (**A**) Western blots showing Cpt1a knockdown (KD) in mouse TRAMPC1 cells. (**B**) Seahorse flux analysis of the Cpt1a-KD cells treated with 25 µM oleic acid as the only carbon source, Holm–Sidak multiple comparisons t-test: * *p* < 0.05; *** *p* < 0.001. (**C**,**D**) Spleen cells from both male (*n* = 4) and female (*n* = 3) C57Bl6 mice were co-cultured with Cpt1A-KD cells (sh2 line) for 48 h and CD4 (**C**) and CD8 (**D**) T-cells were analyzed for PD1 expression by flowcytometry, *t*-test: ** *p* < 0.01.

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
