# Peer review of "Targeting Fat Oxidation in Mouse Prostate Cancer Decreases Tumor Growth and Stimulates Anti-Cancer Immunity"

_ijms, 2020, doi:10.3390/ijms21249660_

Round 1

Reviewer 1 Report

In the manuscript “Targeting fat oxidation in mouse prostate cancer decreases tumor growth and stimulates anti-cancer immunity” the authors explore the role of fatty acid oxidation via CPT1 in suppressing anti-tumor immune infiltration in prostate cancer (PCa). They demonstrate that the lipid oxidation inhibitor ranolazine delivered to mice bearing grafted PCa tumors associated with infiltration of CD8+ Tcells, dendritic cells, and macrophages into tumor tissues and inhibition of tumor growth. Ex vivo co-cultures showed that spleen cells more effectively killed PCa cells after exposure to ranolazine. The specificity of CPT1 in these effects was verified with genetic knockdown. Overall, this study suggests that targeting lipid oxidation in PCa cells may facilitate anti-tumor immunity and could render immune therapies more effective in this cancer type. Importantly, PCa is not known to be widely responsive to immune therapies, and this study sheds light on the role of cancer cell metabolism in influencing the immune milieu.  

This study has several strengths and only minor limitations. Strengths include the use of grafted tumors in wild type mice, the analysis of immune cells by FACS as well as by immunofluorescence, the co-culture experiment with spleen cells and cancer cells, and the combination of pharmacologic and genetic approaches to target CPT1.

Limitations: 1) Are body weight data available from the ranolazine-treated mice to confirm the lack of toxicity? 2) The authors discuss the metabolic phenotype of PCa cells in the introduction; thus, it would be helpful to elaborate on the models they used and how they correspond to the known metabolic profiles of prostate cancer. 3) Could the spleen cell co-culture studies be done in one or two human PCa lines that represent different subtypes to validate the applicability across a range of tumor types? This is only a minor suggestion, and perhaps not possible with the use of mouse spleen cells.

Overall, this work presents the framework for an exciting opportunity to target intrinsic cancer cell metabolism as a means to influence the surrounding immune environment and potentially render tumors more susceptible to anti-immune therapies.

Author Response

Thank you very much for the comments.

Please see attached file with the responses.

Reviewer 2 Report

  • There is no ethics statement in the methods section in relation to the use of scientific animals. Can you please state whether your study was approved by the appropriate ethics committee in your institute. 
  • Essential details that I would expect to see in any paper describing a study in which scientific animals were used are lacking. Please include more details on the mice used in the study- for example type of food, whether they had ad libitum access to food/ water, cage holding conditions etc.
  • There is no mention of female mice in the methods section but there are spleens from female mice used in figure 6. Please provide the details for these animals.
  • It is stated in methods that mice would be euthanised once tumor volume had reached 2CC. Why then are there mice who were given the vehicle control that had tumors over this threshold (Fig 1). This links back to my previous point- what parameters were approved for tumor growth at the beginning of the study?
  • Please use consistent units in the figures and methods sections (e.g. either all CC or mm3). 
  • How did you determine which the drug doses? Were any toxicity studies performed?
  • Could you explain what the individual points in figures 1 and 2 are showing? Are these individual mice? If so, they do not match up with the numbers per group stated in the figure legend. 
  • The legend for figure 2 does not state which statistical test has been performed. The legend also says that this figure shows cells from the mice from figure 1b but they were not given 100mg/Kg Rano- are these actually the mice from figure 1c? This makes this figure confusing. 
  • The legends for figures 3,4, 5 and 6 do not state which statistical tests were used. 
  • How big were the group sizes in figure 4?
  • Were there any differences between the male and female spleen cells (fig 6)?

Author Response

(The authors gave the same response as above.)

Round 2

Reviewer 2 Report

All of my queries have been met.